# Plitidepsin: Mechanisms and Clinical Profile of a Promising Antiviral Agent against COVID-19

**DOI:** 10.3390/jpm11070668

**Published:** 2021-07-16

**Authors:** Michail Papapanou, Eleni Papoutsi, Timoleon Giannakas, Paraskevi Katsaounou

**Affiliations:** 1School of Medicine, National and Kapodistrian University of Athens, 11527 Athens, Greece; mixalhspap13@gmail.com (M.P.); helenapapoutsi@gmail.com (E.P.); tgiannakas@gmail.com (T.G.); 2Society of Junior Doctors, 15123 Athens, Greece; 3Pulmonary and Respiratory Failure Department, First ICU, Evaggelismos Hospital, 10676 Athens, Greece

**Keywords:** plitidepsin, aplidin, COVID-19, SARS-CoV-2, antiviral agents

## Abstract

Current standard treatment of COVID-19 lacks in effective antiviral options. Plitidepsin, a cyclic depsipeptide authorized in Australia for patients with refractory multiple myeloma, has recently emerged as a candidate anti-SARS-CoV-2 agent. The aim of this review was to summarize current knowledge on plitidepsin’s clinical profile, anti-tumour and anti-SARS-CoV-2 mechanisms and correlate this with available or anticipated, preclinical or clinical evidence on the drug’s potential for COVID-19 treatment.PubMed, Scopus, CENTRAL, clinicaltrials.gov, medRxiv and bioRxiv databases were searched.Plitidepsinexerts its anti-tumour and antiviral properties primarily through acting on isoforms of the host cell’s eukaryotic-translation-elongation-factor-1-alpha (eEF1A). Through inhibiting eEF1A and therefore translation of necessary viral proteins, it behaves as a “host-directed” anti-SARS-CoV-2 agent. In respect to its potent anti-SARS-CoV-2 properties, the drug has demonstrated superior ex vivo efficacy compared to other host-directed agents and remdesivir, and it might retain its antiviral effect against the more transmittable B.1.1.7 variant. Its well-studied safety profile, also in combination with dexamethasone, may accelerate its repurposing chances for COVID-19 treatment. Preliminary findings in hospitalized COVID-19 patients, have suggested potential safety and efficacy of plitidepsin, in terms of viral load reduction and clinical resolution. However, the still incomplete understanding of its exact integration into host cell–SARS-CoV-2 interactions, its intravenous administration exclusively purposing it for hospital settings the and precocity of clinical data are currently considered its chief deficits. A phase III trial is being planned to compare the plitidepsin–dexamethasone regimen to the current standard of care only in moderately affected hospitalized patients. Despite plitidepsin’s preclinical efficacy, current clinical evidence is inadequate for its registration in COVID-19 patients.Therefore, multicentre trials on the drug’s efficacy, potentially also studying populations of emerging SARS-CoV-2 lineages, are warranted.

## 1. Introduction

As of March 2020, the World Health Organization (WHO) stressed the need for coordination and direction of global research work towards the development of treatment strategies for coronavirus disease 2019 (COVID-19) [1]. Subsequently, many large clinical trials were carried out, aiming to identify and test the safety and effectiveness of potential therapeutics against the virus [2,3,4,5,6]. Despite the substantial efforts, only a few drugs, such as dexamethasone (an agent with anti-inflammatory rather than antiviral properties) and remdesivir, have been associated with a more favourable disease trajectory [2,4,7,8,9]. Furthermore, a recently published, international randomized trial by the WHO reported that the most widely proposed drugs have had little or no effect on overall mortality of hospitalized patients with COVID-19 [10].

Plitidepsin, a marine cyclic depsipeptide investigated more for its anti-tumour rather than antiviral activity, lately emerged amongst many other candidates as a new therapeutic option for COVID-19 [11,12]. In view of accumulating preclinical data and upcoming clinical trials on patients with COVID-19, we aim to review current knowledge on plitidepsin’sefficacy and safety profiles (also in combination with dexamethasone), illustratively describe its severe acute respiratory syndrome coronavirus-2 (SARS-CoV-2)-related action mechanisms and critically discuss its implications for COVID-19 treatment.

## 2. Materials and Methods

We conducted a comprehensive literature search on plitidepsin, its molecular anti-tumour or antiviral mechanisms, clinical uses and/or current indications, described toxicity profile and preclinical and/or (anticipated, ongoing or completed) clinical studies on the potency and efficacy of the drug against SARS-CoV-2 infection. For this purpose, we searched the PubMed, Scopus, Cochrane Central Register of Controlled Trials (CENTRAL) and clinicaltrials.gov databases, as well as relevant preprint servers (medRxiv, bioRxiv) to also retrieve articles not yet indexed in PubMed. References of the identified articles, along with relevant information on the drug’s authorization by competent authorities all over the world, were also extracted. We used two different search algorithms; one to broadly acquire all current data on plitidepsin (“(plitidepsin OR “plitidepsin” [Appendix A] OR aplidine OR aplidin OR “dehydrodidemnin B”)”) and another to ensure that all evidence correlated with SARS-CoV-2 infection was also obtained(“(plitidepsin OR “plitidepsin” [Appendix A] OR aplidine OR aplidin OR “dehydrodidemnin B”) AND (“COVID-19” [Mesh] OR COVID-19 OR “Coronavirus disease 19” OR “SARS-CoV-2” [Mesh] OR SARS-CoV-2 OR “severe acute respiratory syndrome coronavirus 2”)”). The same algorithms without the Mesh terms were implemented for the databases other than PubMed.Records were initially retrieved on March 23, 2021. Due to the constantly accumulating data on investigated antiviral options, the search was updated on April 15, 2021. The search strategy for each database is extensively provided in Appendix A.

## 3. Mechanisms of Action

### 3.1. Cancer

Plitidepsin is an agent initially studied for its anti-tumour properties (i.e., cell cycle arrest, apoptosis and growth inhibition) [13]. Its antineoplastic activity emerges not only from antiproliferative but also from antiangiogenic (i.e., inhibition of vascular endothelial growth factor secretion) effects [13,14,15,16]. The primary intracellular target of plitidepsin seems to be eukaryotic translation elongation factor 1 alpha 2 (eEF1A2), one of two different isoforms of eukaryotic translation elongation factor 1 (eEF1A). eEF1A2 is responsible for the enzymatic delivery of aminoacyl tRNAs to the ribosome, but also has noncanonical pro-oncogenic activities. Via its inhibition, the drug is engaged in numerous cell actions, such us regulation of oxidative stress, control of unfolded protein degradation by the proteasome, heat shock response and actin building and cytoskeleton reorganization [17,18]. Concerning plitidepsin’s impact on the cell cycle, Alonso et al. [19] presented a dual effect in human melanoma cells, with it being cytostatic at low concentrations and cytotoxic at higher concentrations.

### 3.2. SARS-CoV-2

Ex vivo studies of SARS-CoV-2 and pancoronaviral interactomes identified 332 host proteins interacting with the virus at crucial stages of its life cycle [20,21]. Existent drugs were investigated as “host-directed agents”, targeting these host proteins [21]. Examples include ralimetinib (a p38/MAPK inhibitor) and drugs targeting the eukaryotic translation machinery, like zotatifin (an inhibitor of eukaryotic initiation factor eEIF4A, the partner of eEIF4H that interacts with SARS-CoV-2 Nsp9), plitidepsin and its molecular derivative, ternatin-4 (an eEF1A inhibitor) [22,23].

With regard to plitidepsin, which belongs to the above analysed drug category, White et al. [23] demonstrated that it possesses antiviral activity against SARS-CoV-2 by inhibiting the activity of eEF1A. They proved that the expected anti-SARS-CoV-2 activity of plitidepsin could be mitigated when using a mutated version of eEF1A in 293T cells (A399V mutation), suggesting the factor as a druggable target [23]. eEF1A is used by RNA viruses for mRNA translation by being involved in both the enzymatic delivery of aminoacyl tRNAs to the ribosome and the aminoacylation-dependent tRNA export pathway [24]. It has also been previously identified as an important host factor for the replication of the single-stranded RNA influenza virus, respiratory syncytial virus and certain transmittable coronaviruses [25,26,27,28]. In the case of SARS-CoV-2, and through targeting eEF1A, plitidepsin inhibits the translation of the open reading frames (ORF) ORF1A and ORF1B, leading to reduced production of polyproteins (pp) pp1a and pp1ab, hence conducing to a decreased quantity of replicative non-structural proteins, such as RNA-dependent-RNA-polymerase [29]. It also inhibits the translation of different subgenomic mRNAs, resulting in insufficient production of viral structural and accessory proteins [29]. The lack of necessary viral proteins, such as RNA-dependent-RNA-polymerase, as well as structural proteins simultaneously prevents the virus from generating copies. Figure 1 illustrates the exact mechanism of the drug’s integration into the SARS-CoV-2– host cell interactions [24,30,31].

## 4. Current Uses and Authorization

### 4.1. Multiple Myeloma (MM)

To date, relapsed/refractory MM constitutes the main indication for plitidepsin’s use. Preclinical evidence has suggested that the drug has both in vivo and in vitro anti-MM properties [32]. This activity of plitidepsin was further investigated by clinical trials, solely or in combination with established anti-MM agents, including dexamethasone, in patients with relapsed/refractory MM [33,34,35].

Due to its limited benefit, several concerns were raised as to whether the drug should be authorized; thus, plitidepsin was not universally approved. To date, the combination of plitidepsin with dexamethasone is authorized only in Australia and solely for patients with relapsed/refractory MM after a minimum of three prior treatment regimens [36]. On the contrary, the European Medicines Agency’s (EMA) Committee for Medicinal Products for Human Use refused the authorization of Aplidin^®^ (plitidepsin’s trademark) even for this subset of patients [37]. Nevertheless, as of October 2020, EMA returned the marketing authorization application for Aplidin^®^, and further evaluation of the drug will follow [38].

### 4.2. Leukemia and Lymphomas

Due to its antiproliferative and selective cytotoxic properties in experimental models of human leukaemia cell lines and fresh leukaemia cells, as of 2003, plitidepsin was granted orphan designation by the European Commission for treatment of acute lymphoblastic leukaemia [39]. In this case, the drug should be reassessed in clinical trials before receiving marketing authorization.

Several phase I and II clinical studies have also evaluated plitidepsin in patients with lymphomas; these have estimated safe doses of the drug, also confirming feasibility of its combination with other anti-tumour agents [40,41,42].

### 4.3. Melanoma

According to preclinical evidence, plitidepsinappears to act as an antiproliferative agent in melanoma cell lines while presenting synergistic activity with dacarbazine, a drug used for treating metastatic melanoma [19]. Nevertheless, studies carried out to evaluate the efficacy and safety of plitidepsin in patients with locally advanced or metastatic malignant melanoma revealed only a limited clinical benefit [43,44]. On this basis, the drug was not included in 2019 European Society for Medical Oncology (ESMO) Clinical Practice Guidelines for treatment of this malignancy [45].

## 5. Safety Profile

Plitidepsin’s safety profile has been extensively studied over both phase I and II/III trials, with the drug presenting mostly transient and tolerated adverse events. Myalgia, alanine aminotransferase/aspartate aminotransferase and creatine phosphokinase increase, as well as fatigue, nausea and vomiting, constituted the most common dose-limiting toxicities in phase I studies [34,41,46,47,48]. Regarding phase II/III studies, the same adverse events were reported, along with mild to moderate hematologic abnormalities, such as anaemia and thrombocytopenia [33,35,43].A hypersensitivity reaction was also observed; the event was well-tolerated with the exception of one patient that was withdrawn from a trial due to a grade 4 hypersensitivity reaction and hypotension [43]. 

A particularly well-studied drug combination among patients with relapsed and refractory multiple myeloma consisted of plitidepsin with dexamethasone [33,34,35]. Even though the combination was associated with a mildly increased incidence of muscular events and creatine phosphokinase elevations, the safety profile is altogether acceptable [33]. Notably, plitidepsin with dexamethasone was related to less hepatic enzymes abnormalities compared to plitidepsin alone [33], while a study assessing the combination of plitidepsin with dexamethasone and bortezomib reported no dose-limiting adverse events [34]. Consequently, the drug is considered safe and well-tolerated, both as a monotherapy and as a combination with dexamethasone; hence, it is already authorized for use in patients with relapsed/refractory MM in Australia [36].

## 6. Implications for COVID-19 Treatment

Both ex vivo and in vivo preclinical data on plitidepsin’s antiviral activity against SARS-CoV-2 have recently emerged [23]. Its antiviral effect in infected Vero E6 cells has surpassed that of other currently studied host-directed agents (i.e., termatin-4, zotatifin) [23]. Compared to the current standard of care, remdesivir, plitidepsin has been proven more potent in reducing the expression of the viral structural protein N in Vero E6 cells and 27.5 times more powerful in inhibiting SARS-CoV-2 replication in the hACE2-293T human cell line [23]. Among 72 in vitro tested potentially antiviral drugs, plitidepsin was the only clinically approved drug exhibiting nanomolar efficacy (expressed as IC50) against SARS-CoV-2 replication and subsequent cytopathic effects [49]. In vivo, the drug has decreased lung virus titres and lung pathology of infected mice to a comparable extent to remdesivir [23]. Considering that emerging (and potentially remdesivir-resistant) variants may outweigh earlier strains, an advantage of plitidepsin is that it appears to preserve its antiviral activity against the more transmittable and likely more deadly (though data on increase in severity or death are contradictory) B.1.1.7 variant, as demonstrated by the results of a preprint study [22,50,51,52,53]. This property may be attributed to the drug targeting host’s proteins vital for the virus life cycle, yet less amenable to mutations than viral proteins, as described above [22]. Similar host’s mechanisms have been located as essential for replication of other respiratory viruses, including influenza, respiratory syncytial virus and other transmittable coronaviruses, implying a potent utility of such agents for combating future coronavirus or other viral outbreaks [25,26,27,28]. 

What currently further distinguishes plitidepsin from other candidate host-directed anti-SARS-CoV-2 therapies is the more adequate comprehension of its bioavailability and safety profiles, even in cases of co-administration with dexamethasone, which is part of the current standard of care of hospitalized patients with COVID-19, primarily through available MM trials [33,34,35]. Such knowledge will greatly accelerate the drug’s testing in infected hospitalized patients requiring oxygen supplementation or invasive mechanical ventilation [4]. The drug’s safety and efficacy have also been preliminarily evaluated in the specific setting of COVID-19 patients needing hospital admission [54]. In this proof-of-concept trial, three cohorts of patients intravenously (IV) received three different doses (1.5 mg, 2.0 mg or 2.5 mg per day) of plitidepsin over three consecutive days post-admission [54]. The protocol further predefined an obligatory minimum of hospitalization for 7 days and surveillance for adverse events for a timeframe of 31 days after admission [54]. Except being well-tolerated and without any serious adverse events being observed, plitidepsin reduced participants’ viral load by 50% and 70% on hospitalization days 7 and 15, respectively, and it concomitantly led to 38% and 81% of patients being discharged by days 8 and 15, respectively [54,55]. Viral load reduction was significantly associated with clinical resolution of pneumonia and decrease in C-reactive protein levels [54,55]. On their visit on day 30, no patients presented with symptoms compatible with COVID-19 [54,55]. 

When the drug’s characteristics are carefully examined, certain deficits might be identified. Since plitidepsin’s marketing authorization is so far limited to the narrow clinical setting of patients with relapsed/refractory MM and only in Australia, the drug is not widely incorporated into daily clinical practice and is therefore not broadly known by clinicians.Affecting multiple cell cycle-related molecular pathways is accompanied by a specific toxicity profile [33,35,43]. This toxicity profile encompasses more common dose-limiting adverse events, such as fatigue, nausea, vomiting, anemia and thrombocytopenia, to rare cases of hypersensitivity [33,35,43]. Interestingly, the administered IV antitumor doses of the drug (including the dose authorized for administration in Australia) (5 mg/m^2^) were higher than the ones planned for hospitalized COVID-19 patients (a maximum of 2.5 mg per day for three days post-admission) [33,35,43,54]. Following the examples of hydroxychloroquine, lopinavir–ritonavir and remdesivir, plitidepsin remains a repurposed drug with its mechanisms being studied chiefly in the context of its anti-tumour properties [56]. Therefore, plitidepsin’s specificity against SARS-CoV-2, as well as the exact host–virus interactive mechanisms with which the agent interferes, should be further elucidated. To facilitate this purpose, Martinez suggested the generation of tissue cultures resistant to the drug [55]. Although the drug’s host-directed mechanisms may also imply activity against different SARS-CoV-2 variants, current evidence supporting such an effect derives only from a preprint preclinical study and should therefore be considered anecdotal [22]. Accounting for its intravenous route of administration, the drug is also excluded from a community prophylactic use in mildly affected individuals, a setting for which beneficial antiviral options have not been discovered yet [55]. Though several steps are required before developing oral analogues, plitidepsin is currently clearly purposed for hospitalized patients with moderate severity of infection [54,57]. 

Although encouraging enough, current evidence on plitidepsin’s utility for treating SARS-CoV-2 infection has been obtained by preclinical in vivo or ex vivo studies and a single multicentre phase I/II trial with 46 participants, designed and conducted by the PharmaMar company. In absence of remarkable antiviral weapons for the fight against SARS-CoV-2, current limited evidence definitely merits further carefully designed, multicentre, randomized controlled trials that will either establish or disprove the safety and efficacy of plitidepsin, also in comparison with the standard of care. As the landscape of dominant strains is constantly changing, it would be prudent to later enrol individuals infected by the emerging variants and separately study the drug’s activity also in these population subgroups [50]. In the context of all these evolutions and in order to examine whether the drug possesses a true therapeutic benefit for hospitalized patients with COVID-19 of moderate severity, the PharmaMar company is planning to initiate the multicentre, phase III NEPTUNO trial (NCT04784559) [57]. According to the trial’s protocol, participants will be randomized in a 1:1:1 ratio to receive either IV plitidepsin at 1.5 mg per day combined with dexamethasone, IV plitidepsin at 2.5 mg per day combined with dexamethasone or dexamethasone alone, with remdesivir being contextually added to the regimen (as per local treatment guidelines) at IV 200 mg on day 1 followed by IV 100 mg per day on days 2 to 5 after admission [57]. In the first two arms, and in adherence to the previous phase I/II trial’s protocol, plitidepsin will be administered in two of the formerly tested dosages and only over the first three consecutive hospitalization days [54,57]. Dexamethasone in these arms will also be given at 8 mg per day IV on days 1 to 3, followed by 6 mg per day orally or IV (depending on the physician’s judgment of patient’s condition) from day 4 and up to day 10 [57]. Due to the mandatory need for direct antiviral options, as well as the drug’s preclinical efficacy and host-directed mechanisms analysed above, the results of the trial are anticipated with great scientific interest and may form an entirely different approach in the treatment of COVID-19 patients. However, the trial’s design reinforces the assumption that the drug is aimed at hospitalized patients and only those with disease of moderate severity. Such a design excludes, at this point, severely affected individuals, a population for which effective antiviral options may be of considerable benefit.Therefore, more research is needed as this trial, even in case it demonstrates significant clinical efficacy against the enrolled group of patients, constitutes just the first “crash-test” for plitidepsin.

Despite plitidepsin’s preclinical efficacy against SARS-CoV-2, clinical evidence is currently inadequate for its registration in COVID-19 patients. Even in case clinical efficacy of the drug against moderately affected COVID-19 patients is demonstrated by the NEPTUNO trial, many issues remain to be addressed in the future, including the drug’s effectiveness against different SARS-CoV-2 variants and its efficacy when administered in severely affected hospitalized patients.As the crucial research for antiviral options against SARS-CoV-2 is progressing, evidence on all emerging and promising candidate agents should be carefully and constantly assessed and updated. Moreover, as SARS-CoV-2 may be part of our daily routine the following years, it is desirable to have pharmaceutical options for all patients and variants and for all levels of severity of the disease. In this light, plitidepsin is likely to gain a foothold in patients of moderate severity and in some SARS-CoV-2 strains.

## 7. Conclusions

Plitidepsin, a drug currently authorized only in Australia for patients with refractory multiple myeloma, has exhibited anti-SARS-CoV-2 properties, through inhibiting the elongation factor eEF1A, a component of the eukaryotic host cell’s translation machinery. On the basis of accumulating preclinical and only preliminary clinical data, plitidepsin currently represents a promising repurposed candidate drug against COVID-19 requiring hospitalization.However, current clinical evidence is inadequate for plitidepsin’s use in COVID-19 patients, while several issues, such as its efficacy against variants and its purposing only for moderately affected hospitalized patients, remain to be addressed. In an urgent necessity for more antiviral agents, that also retain activity against the constantly spreading SARS-CoV-2 variants and as pandemic conditions are changing, carefully designed, multicentrerandomized controlled trials, potentially further studying separate subgroups of patients infected by new strains, are warranted.Thus, plitidepsin’s role may be readdressed in a subcategory of COVID-19 patients that might benefit from the drug, satisfying a personalized approach of these patients in the future.

## Figures and Tables

**Figure 1 jpm-11-00668-f001:**
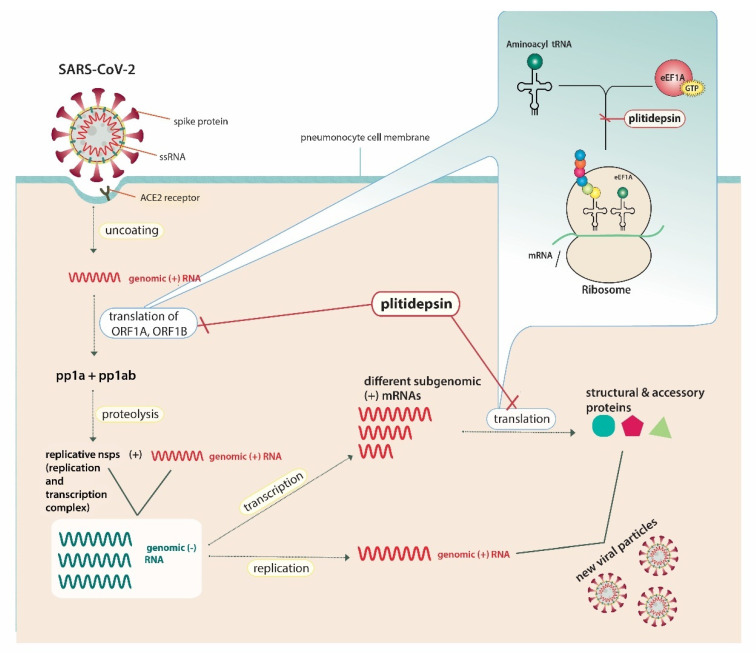
Plitidepsin’s host-directed anti-SARS-CoV-2 action mechanisms.SARS-CoV-2 possesses a single-stranded RNA (ssRNA) genome. The angiotensin converting enzyme-2 (ACE2) receptor is identified as the cell-surface receptor of SARS-CoV-2. Specific spike protein interactions with ACE2 receptors promote viral fusion with the cellular membrane. After entry, uncoating of the viral genomic RNA is followed by the translation of two large open reading frames (ORF), ORF1A and ORF1B. The resulting polyproteins, pp1a and pp1ab, are proteolyzed into non-structural proteins (nsps) that form the viral replication and transcription complex. This complex includes, amongst others, RNA-processing and RNA-modifying enzymes, such as RNA-dependent-RNA-polymerase, and drives the production of negative-sense RNAs ((−) RNAs). In general, the positive-sense genome can act as messenger RNA (mRNA) and can be directly translated into viral proteins, whereas negative-sense RNA is converted (via RNA-dependent-RNA-polymerase) into positive-sense RNA in order to be translated. Genomic RNA contains the necessary RNA regions required for genome replication and translation. During replication, full-length (−) RNA copies of the genome (genomic (−) RNAs) are used as templates for genomic (+) RNAs. During transcription, various subgenomic RNAs are produced through discontinuous transcription, where subgenomic (−) RNAs are synthesized by combining varying lengths of the 3′ end of the genome with the 5′ leader sequence necessary for translation. Subgenomic (−) RNAs are then transcribed into subgenomic (+) mRNAs. Resulting structural and accessory viral proteins are combined with genomic (+) RNAs to produce new viral particles, which will be secreted from the infected pneumonocyte by exocytosis. Through targeting the host cell’s eukaryotic translation elongation factor (eEF1A), plitidepsin inhibits the host-mediated translation of ORF1A, ORF1B and subgenomic mRNAs, leading to decreased production of viral pp1a and pp1ab andnsps, including RNA-dependent-RNA-polymerase, as well as structural and accessory proteins. Abbreviations: SARS-CoV-2, severe acute respiratory syndrome coronavirus-2; ssRNA, single-stranded RNA; ACE2, angiotensin converting enzyme-2; ORF, open reading frame; mRNA, messenger ribonucleic acid; tRNA, transfer ribonucleic acid; eEF1A, eukaryotic translation elongation factor 1; GTP, guanosine triphosphate; pp, polyprotein.

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
