# Peer review of "Plitidepsin: Mechanisms and Clinical Profile of a Promising Antiviral Agent against COVID-19"

_jpm, 2021, doi:10.3390/jpm11070668_

Round 1

Reviewer 1 Report

I liked your review , comprehensive and useful.Οnly:

1)I would like the reasons for the rejection of the medicine to be mentioned in great detail.

2) Your personal opinion for the drug

Author Response

Point 1: “I liked your review, comprehensive and useful.”

Authors’ reply: Thank you for dedicating valuable time and effort to review our manuscript.

Point 2:I would like the reasons for the rejection of the medicine to be mentioned in great detail.”

Authors’ reply: Thank you for your constructive remark. Apart from the previously mentioned toxicity profile, intravenous route of administration, lack of knowledge related to exact mechanisms, and precocity of clinical data, we have enriched our discussion on the drug’s deficits, describing them to a greater extent, as suggested:

“Since plitidepsin’s marketing authorization is so far limited to the narrow clinical setting of patients with relapsed/refractory MM and only in Australia, the drug is not widely incorporated into daily clinical practice and therefore not broadly known by clinicians.” (Page 6, lines 241-244)

“Although the drug’s host-directed mechanisms may also imply activity against different SARS-CoV-2 variants, current evidence supporting such an effect derives only from a preprint preclinical study and should therefore be considered anecdotal [22].” (Page 6, lines 256-259)

“However, the trial’s design reinforces the assumption that the drug is aimed at hospitalized patients and only those with disease of moderate severity. Such a design excludes, at this point, severely affected individuals, a population for which effective antiviral options may be of considerable benefit. Therefore, more research is needed as this trial, even in case it demonstrates significant clinical efficacy against the enrolled group of patients, constitutes just the first “crash-test” for plitidepsin.”(Page 7, lines 290-296)

Point 3:“Your personal opinion for the drug.”

Authors’ reply:Thank you for your insightful suggestion.Taking into account all existing evidence, we have attempted to summarize our current opinion for the drug in the following statement:“Despite plitidepsin’s preclinical efficacy against SARS-CoV-2, clinical evidence is currently inadequate for its registration in COVID-19 patients. Even in case clinical efficacy of the drug against moderately affected COVID-19 patients is demonstrated by the NEPTUNO trial, many issues remain to be addressed in the future, including the drug’s effectiveness against different SARS-CoV-2 variants and its efficacy when administered in severely affected hospitalized patients.” (Page 7, lines 297-304)A relevant statement has been added in both text and graphical versions of the abstract, so that our current conclusion is more clearly depicted: “Despite plitidepsin’s preclinical efficacy, current clinical evidence is inadequate for its registration in COVID-19 patients.” (Page 1, lines 29-30)Another relevant statement has been added in the “Conclusions” section: “However, current clinical evidence is inadequate for plitidepsin’s use in COVID-19 patients, while several issues, such as its efficacy against variants and its purposing only for moderately affected hospitalized patients, remain to be addressed.” (Page 7, lines 311-314)Finally, we have added a statement referring to the mandatory need for constant evaluation of evidence on emerging promising anti-SARS-CoV-2 agents: “As the crucial research for antiviral options against SARS-CoV-2 is progressing, evidence on all emerging and promising candidate agents should be carefully and constantly assessed and updated. Moreover as SARS-CoV-2 may be part of our daily routine the following years it is desirable to have pharmaceutical options for all patients and variants and for all levels of severity of the disease. In this light, Plitidepsin is likely to gain a foothold in patients of moderate severity and in some SARS-CoV-2 strains. ”(Page 7, lines 302-308)

Reviewer 2 Report

This manuscript reviewed the drug plitidepsin (aplidin) as a therapeutic option for COVID-19 patients. Plitidepsin is originally used for the treatment of multiple myeloma (non-COVID-19 patients). The authors reviewed plitidepsin in four different aspects:

  • mechanisms action of plitidepsin among non-COVID-19 patients and COVID-19 patients (section 3, lines 72-140)
  • current uses of plitidepsin in non-COVID-19 patients (section 4, lines 141-173)
  • safety profile of plitidepsin in non-COVID-19 patients (section 5, lines 174-194)
  • plitidepsin responses against SARS-CoV-2 or COVID-19 patients (section 6, lines 195-275)

The methodology to perform this review was clear.

Regarding the responses of plitidepsin against SARS-CoV-2, the authors cited the findings from the three publications [22, 23, 49]. These findings were published by two different research groups. One group [22, 23] showed that plitidepsin can: (1) reduce the expression of SARS-CoV-2 N protein in Vero E6 cells, (2) inhibit SARS-CoV-2 replication in hACE2-293T cells, (3) decrease lung virus titers and lung pathology of infected mice, (4) share similar antiviral activity against both the early-lineage and B.1.1.7 variant SARS-CoV-2. Another group [49] found that plitidepsin could attain minimum IC50 among the 72 antiviral compounds tested.

Regarding the responses of plitidepsin against COVID-19 patients, the authors cited the discussions from a commentary article [55]. This commentary article summarized the findings of the two references [22, 23] and discussed the possible further studies.

There is limited information regarding the usage of plitidepsin against COVID-19 patients. The present manuscript only rephrased the previous findings of 22, 23, 49. The commentary article [55] has already discussed the views mentioned by the authors. In light of the information received from this review, it produces little or no insightful information.

Author Response

Point 1:“This manuscript reviewed the drug plitidepsin (aplidin) as a therapeutic option for COVID-19 patients. Plitidepsin is originally used for the treatment of multiple myeloma (non-COVID-19 patients). The authors reviewed plitidepsin in four different aspects:

  • mechanisms action of plitidepsin among non-COVID-19 patients and COVID-19 patients (section 3, lines 72-140)
  • current uses of plitidepsin in non-COVID-19 patients (section 4, lines 141-173)
  • safety profile of plitidepsin in non-COVID-19 patients (section 5, lines 174-194)
  • plitidepsin responses against SARS-CoV-2 or COVID-19 patients (section 6, lines 195-275)

The methodology to perform this review was clear.

Regarding the responses of plitidepsin against SARS-CoV-2, the authors cited the findings from the three publications [22, 23, 49]. These findings were published by two different research groups. One group [22, 23] showed that plitidepsin can: (1) reduce the expression of SARS-CoV-2 N protein in Vero E6 cells, (2) inhibit SARS-CoV-2 replication in hACE2-293T cells, (3) decrease lung virus titers and lung pathology of infected mice, (4) share similar antiviral activity against both the early-lineage and B.1.1.7 variant SARS-CoV-2. Another group [49] found that plitidepsin could attain minimum IC50 among the 72 antiviral compounds tested.

Regarding the responses of plitidepsin against COVID-19 patients, the authors cited the discussions from a commentary article [55]. This commentary article summarized the findings of the two references [22, 23] and discussed the possible further studies.

There is limited information regarding the usage of plitidepsin against COVID-19 patients. The present manuscript only rephrased the previous findings of 22, 23, 49. The commentary article [55] has already discussed the views mentioned by the authors. In light of the information received from this review, it produces little or no insightful information.”

Authors’ reply: Thank you for dedicating valuable time and effort to review our manuscript. Current evidence on plitidepsin’s potential anti-SARS-CoV-2 effect derives from only few and specific research groups. In order to inform the readers about a drug not broadly known or used by clinicians (since the drug has received marketing authorization only for the very specific clinical setting of patients with relapsed or refractory multiple myeloma and only in Australia), we adopted a more well-rounded approach and attempted to review not only plitidepsin’s SARS-CoV-2 – related research, but also its current uses, further authorization details and safety profile (including a summary of reported adverse events by pre-pandemic literature). We have also attempted to provide a comprehensive illustration of the drug’s complex and not easily perceptible host-directed action mechanisms. In this way, we hoped to depict how inhibition of certain eukaryotic translation pathways may lead to an anti-SARS-CoV-2 effect. In order to more accurately inform the readership about the goals of this review, we have modified the title so that it now reads as follows: “Plitidepsin: mechanisms and clinical profile of a promising antiviral agent against COVID-19”.

Since our research was also COVID-19 – orientated, we focused on the reported clinical uses and safety profile of plitidepsin’s coadministration with dexamethasone. Through examining relevant pre-pandemic literature and in view of potential upcoming trials in hospitalized COVID-9 patients, we aimed at further exploring the extent to which this combination of drugs has already been tested and/or used in a clinical setting. Our final goal was to address future research perspectives through examining whether clinical trials are being designed to further assess plitidepsin’s safety and efficacy against COVID-19 patients. We therefore analyzed the design of the anticipated phase III NEPTUNO trial. By that means, we hoped to serve a dual purpose: examine whether the trial’s design adheres to the previously implemented protocols, and elucidate the exact clinical setting for which the drug is currently being purposed.

In view of our attempt to cover a broader spectrum of evidence related to the drug and in accordance with your constructive remarks, we have expanded on our discussion on perceived deficits of the drug’s use against COVID-19 patients and have added comments that will hopefully capture the aspects of our non-COVID-19-related research in relation to the drug’s implications for COVID-19 treatment, or will provide some new insight for future research. The added comments as well as their exact location in text are provided below:

“Since plitidepsin’s marketing authorization is so far limited to the narrow clinical setting of patients with relapsed/refractory MM and only in Australia, the drug is not widely incorporated into daily clinical practice and is therefore not broadly known by clinicians.” (Page 6, lines 241-244)

“This toxicity profile encompasses more common dose-limiting adverse events such as fatigue, nausea, vomiting, anemia and thrombocytopenia to rare cases of hypersensitivity [33,35,43]. Interestingly, the administered IV antitumor doses of the drug (including the dose authorized for administration in Australia) (5 mg/m2) were higher than the ones planned for hospitalized COVID-19 patients (a maximum of 2.5 mg per day for three days post-admission) [33,35,43,54].” (Page 6, lines 245-250)

“Although the drug’s host-directed mechanisms may also imply activity against different SARS-CoV-2 variants, current evidence supporting such an effect derives only from a preprint preclinical study and should therefore be considered anecdotal [22].” (Page 6, lines 256-259)

“However, the trial’s design reinforces the assumption that the drug is aimed at hospitalized patients and only those with disease of moderate severity. Such a design excludes, at this point, severely affected individuals, a population for which effective antiviral options may be of considerable benefit. Therefore, more research is needed as this trial, even in case it demonstrates significant clinical efficacy against the enrolled group of patients, constitutes just the first “crash-test” for plitidepsin.”(Page 7, lines 290-296)

Despite plitidepsin’s preclinical efficacy against SARS-CoV-2, clinical evidence is currently inadequate for its registration in COVID-19 patients. Even in case clinical efficacy of the drug against moderately affected COVID-19 patients is demonstrated by the NEPTUNO trial, many issues remain to be addressed in the future, including the drug’s effectiveness against different SARS-CoV-2 variants and its efficacy when administered in severely affected hospitalized patients. As the crucial research for antiviral options against SARS-CoV-2 is progressing, evidence on all emerging and promising candidate agents should be carefully and constantly assessed and updated.” (Page 7, lines 297-304)

Although some of the information provided in our review has been already discussed on the commentary article [55] we tried to additionally deepen in the mechanism of action (figure 1, graphical abstract) and to critically comment in the results of trials so far. Thus, as new variants are introduced and pandemic conditions are changing, plitidepsin, we aim in helping determining the subcategory of Covid-19 patients who are likely to benefit from the drug, satisfying a more personalized treatment of these patients. This is reflected now in the revised conclusion

In an urgent necessity for more antiviral agents, that also retain activity against the constantly spreading SARS-CoV-2 variants and as pandemic conditions are changing, carefully designed, multicentre randomized controlled trials potentially further studying separate subgroups of patients infected by new strains, are warranted. Thus, plitidepsin’s role may be readdressed in a subcategory of Covid-19 patients that might benefit from the drug, satisfying a personalized approach of these patients in the future.”

Reviewer 3 Report

Recently, it has been reported that anti-cancer drug plitidepsin has a potent anti-COVID19 virus activity (PMID: 33495306). The mechanisms/implications of this drug have recently been reviewed/published (PMID: 33558296). The present review explains the updates and mechanisms of action of this drug. Also, it presents a more comprehensive study on the drug uses and side effects.

Minor comments

Line 200. “virus’s” should read “viral”

Line 211. “virus’s” should read “virus”

Author Response

Point 1:“Recently, it has been reported that anti-cancer drug plitidepsin has a potent anti-COVID19 virus activity (PMID: 33495306). The mechanisms/implications of this drug have recently been reviewed/published (PMID: 33558296). The present review explains the updates and mechanisms of action of this drug. Also, it presents a more comprehensive study on the drug uses and side effects.”

Authors’ reply:Thank you for dedicating valuable time and effort to review our manuscript.

Point 2:Minor comments. Line 200. “virus’s” should read “viral”; Line 211. “virus’s” should read “virus”.”

Authors’ reply: Thank you for your observation. We have replaced each “virus’s” with the correct terms “viral”(page 5, line 204) and “virus” (page 5, line 215).

Round 2

Reviewer 2 Report

The authors have addressed my queries and revised the manuscript accordingly.